# Association between Serum 8-Iso-Prostaglandin F2α as an Oxidative Stress Marker and Immunological Markers in a Cohort of Preeclampsia Patients

**DOI:** 10.3390/life13122242

**Published:** 2023-11-22

**Authors:** Lidia Boldeanu, Constantin-Cristian Văduva, Daniel Cosmin Caragea, Marius Bogdan Novac, Mariana Manasia, Isabela Siloși, Maria Magdalena Manolea, Mihail Virgil Boldeanu, Anda Lorena Dijmărescu

**Affiliations:** 1Department of Microbiology, Faculty of Medicine, University of Medicine and Pharmacy of Craiova, 200349 Craiova, Romania; barulidia@yahoo.com; 2Department of Obstetrics and Gynecology, Faculty of Medicine, University of Medicine and Pharmacy of Craiova, 200349 Craiova, Romania; magdalena.manolea@umfcv.ro (M.M.M.); lorenadijmarescu@yahoo.com (A.L.D.); 3Department of Nephrology, Faculty of Medicine, University of Medicine and Pharmacy of Craiova, 200349 Craiova, Romania; caragea.daniel87@yahoo.com; 4Department of Anesthesiology and Intensive Care, Faculty of Medicine, University of Medicine and Pharmacy of Craiova, 200349 Craiova, Romania; 5Doctoral School, University of Medicine and Pharmacy of Craiova, 200349 Craiova, Romania; daria_med@yahoo.com; 6Department of Immunology, Faculty of Medicine, University of Medicine and Pharmacy of Craiova, 200349 Craiova, Romania; isabela_silosi@yahoo.com; 7Medico Science SRL—Stem Cell Bank Unit, 200690 Craiova, Romania; mihail.boldeanu@umfcv.ro

**Keywords:** 8-iso-prostaglandin F2-alpha, oxidative stress, immunological markers, preeclampsia

## Abstract

Background: We aimed to analyze the presence and clinical use of serum 8-iso-prostaglandin F2-alpha (8-iso-PGF2α) as an oxidative stress marker and some inflammatory status biomarkers (tumor necrosis factor-alpha (TNF-α), interleukin 6 (IL-6), IL-10, high-sensitivity C-reactive protein (hs-CRP), and pentraxin-3 (PTX3)) for patients with preeclampsia (PE). Methods: Sixty pregnant women, including thirty diagnosed with PE and thirty who were healthy (NP), were included in this study. For the assessment of serum levels of biomarkers, we used the Enzyme-Linked Immunosorbent Assay (ELISA) technique. Results: Our preliminary study showed that the expression level of serum 8-iso-PGF2α in the PE group was higher than in the PE after delivery (PE-AD) group (742.00 vs. 324.00 pg/mL, *p* < 0.0001). Groups of preeclamptic patients (PE + PE-AD) expressed significantly elevated levels for all of the assessed inflammatory mediators as compared to NP. Significant strong positive correlations with 8-iso-PGF2α levels were found for systolic blood pressure (SBP), and TNF-α (Spearman’s rho = 0.622, *p*-value = 0.020 and rho = 0.645, *p*-value = 0.002, respectively). Our study demonstrates that 8-iso-PGF2α and PTX3 have the greatest diagnostic value for pregnant women with PE. Conclusions: 8-iso-PGF2α and PTX3 can be used as independent predictor factors, along with already-known cytokines, that could represent a prophylactic way to help clinicians identify or predict which pregnant women will develop PE.

## 1. Introduction

Preeclampsia (PE) is a serious complication of pregnancy and one of the leading causes of maternal and neonatal mortality and morbidity in the world. PE affects approximately 2–8% of pregnancies worldwide [1,2].

The diagnosis and prediction of PE are currently based on non-specific clinical signs such as hypertension and proteinuria. More recently, the existence of target organ damage—such as impaired liver function, maternal neurological or hematological problems, and renal insufficiency—in the absence of proteinuria has also been classified as PE [1,3].

Despite the numerous studies already carried out up to now, the etiology of PE remains to be elucidated. Thus, studies have detected abnormal levels of cytokines and angiogenic factors in the placenta, as well as in the maternal circulation, suggesting that these immunological factors play an important role in the development of PE [4,5,6,7,8]. It is unclear how these cytokines relate to the etiology of PE, and the data that are now available on them are contentious, notwithstanding these studies. Whether a particular cytokine’s concentration or ratio is the cause of PE’s pathophysiology is likewise unclear.

Studies have shown that, unlike normal pregnancies, in pregnancies associated with PE, there is an inappropriate immune response, characterized by an alteration of immune cells and activated cytokines [9,10]. Insufficient trophoblast invasion in PE has been observed, leading to placental ischemia, which produces an imbalance in immune function with the appearance of chronic and uncontrolled inflammation, an environment similar to an autoimmune disease [11,12,13]. The immune imbalance consists of the increase in pro-inflammatory immune cells and cytokines, such as tumor necrosis factor-alpha (TNF-α), interferon-γ (IFN-γ), interleukin 1β (IL-1β), and IL-12 [6,14], synthesized by the T helper 1 (Th1) type of lymphocytes, and the decrease in regulatory immune cells and cytokines, such as IL-10, synthesized by the Th2 type, in the placenta and maternal circulation [10,15,16,17,18,19,20,21].

Studies have shown that in the pathogenesis of preeclampsia, the paternal immune response is important in addition to the altered maternal immune response [22]. In couples with azoospermia, in vitro fertilization and intracytoplasmic sperm injection procedures are performed with sperm obtained by means of surgical testicular retrieval [23,24]. Several studies have shown that preeclampsia is greatly increased in pregnancies resulting from these cases [25,26,27].

This immune imbalance is also thought to participate in the overall pathophysiology associated with PE, with the production of reactive oxygen species (ROS) [28,29], the overexpression of endothelin-1 [30,31], and the B cell response. The overexpression of B lymphocytes has been indicated by the production of autoantibodies to the angiotensin II (AngII) type 1 receptor (AT1-AA), which culminates in the development of hypertension during pregnancy [14,32,33,34]. When AT1-AA binds to the AT1 receptor, pro-inflammatory transcription factors in trophoblasts and vascular cells are activated, which can cause the generation of ROS and TNF-α [35,36].

A state of extreme imbalance between the body’s antioxidant defense systems and the increased production of free radicals, such as reactive nitrogen species (RNS) or ROS, is known as oxidative stress [37,38,39,40].

Several tests are available to estimate peroxidation and oxidative damage to lipids that include the measurement of conjugated dienes, isoprostanes, serum malondialdehyde (MDA) levels, serum 4-hydroxy-2-nonenal (4-HNE), or hydroperoxides. Lipid peroxidation is most often measured using MDA and 8-iso-prostaglandin F2-alpha (8-iso-PGF2α) [41,42,43]. Since free radical activity produces persistent products of arachidonic acid peroxidation, isoprostanes are dependable indicators of oxidative stress and can be used to measure lipid peroxidation instead of MDA [44]. 8-iso-PGF2α is one of the isoprostanes that are easily detectable, persistent in bodily fluids, unaffected by food, and controlled by endogenous antioxidants [45,46,47]. A study conducted in Japan revealed significantly increased F2-isoprostane levels at the end of pregnancy compared to non-pregnancy [48].

We began by selecting the research topic based on the observation that there is insufficient information on the role of oxidative stress development as indicated by 8-iso-PGF2α in PE, regarding the follow-up of women after childbirth, as well as the supposed association between 8-iso-PGF2α and immunological markers, such as TNF-α, IL-6, IL-10, high-sensitivity C-reactive protein (hs-CRP), a type of short pentraxin, and pentraxin-3 (PTX3), as a member of the long pentraxin branch. PTX3 is involved in both inflammation and innate immunity and deficiency in mice [49]. Recent studies have shown that pregnancy is associated with increased serum levels of PTX3 compared to non-pregnant women [50], and higher levels of PTX3 have been observed to be associated with PE. PTX3 has been suggested as a valuable early indicator for PE [51,52,53,54,55].

Finding out if these indicators can enhance our ability to evaluate patients in real-time and offer potential guidance on a personalized treatment plan is the primary goal. We may be able to predict the course, prognosis, and quality of life of these individuals in the future with greater accuracy if we can identify specific serum biomarkers that are connected to disease activity. During our study, we set out to answer the following questions:(1)Can the interrelation between serum levels of 8-iso-PGF2α and TNF-α, IL-6, IL-10, hs-CRP, and PTX3 modulate the appearance of PE in the second half of pregnancy (22–28 weeks) compared to that of pregnant women without PE?(2)Does the assessment of serum 8-iso-PGF2α levels have the diagnostic accuracy (using receiver-operator characteristic curve analysis, ROC) required to discriminate between the PE and normal patients?

## 2. Materials and Methods

### 2.1. Study Design

For the implementation of this study, the authors obtained permission from the Ethical Research Committees of the University of Medicine and Pharmacy of Craiova, Romania (no. 139/29 June 2022). The study respected and followed all standards in terms of medical ethics, as recommended in the Helsinki Declaration of 1975, updated in 2008.

The study was designed in three main phases: (i) baseline—initial inflammatory and oxidative stress status assessment, in the second trimester of pregnancy in patients with PE, compared to that of pregnant women without PE, serum sampling (for the quantitative analysis of 8-iso-PGF2α, TNF-α, IL-6, IL-10, hs-CRP and PTX3; (ii) monitoring pregnancies to term with the delivery of healthy term infant; (iii) recall—reassessment of inflammatory and oxidative stress status, in the postpartum period (three months after delivery), and resampling of serum samples (to verify the hypothesis of a decrease in their expression to a level comparable to the normal one). The resulting data were subsequently used for statistical analysis.

After being informed about the study in advance, each patient signed an informed consent form.

### 2.2. Patient Selection

This prospective study was conducted in the Department of Gynecology of the Filantropia Municipal Clinical Hospital, Craiova, Dolj County, Romania, between September 2022 and March 2023.

We included 100 pregnant women in this study, both nulliparous and multiparous, normotensive, aged between 21 and 37 years, having a single pregnancy at 18–20 weeks of gestation. At visit 1 (18–20 weeks of gestation), they were evaluated clinically, sociodemographically, and obstetrically. Pregnant women were followed from 21 weeks of gestation until the time of PE diagnosis, delivery, and three months after delivery. Of the 100 pregnant women, 60 completed the study and were included in the final evaluation, while 40 pregnant women were lost to follow-up.

At visit 2 (24–28 weeks of gestation), 30 of the 60 pregnant women developed PE and were considered as PE patients (PE group), while the other 30 pregnant women were normotensive and were considered as the control (NP group).

A total of 40 women were lost to follow-up due to pregnancy complications, diabetes mellitus (*n* = 10), chronic kidney disease (*n* = 8), systemic lupus erythematosus (*n* = 2), clinically evident infections (*n* = 6), liver disease (*n* = 4), unwillingness to continue (*n* = 7), and relocation (*n* = 3).

At visit 3 (three months after delivery), the patients were clinically re-evaluated (PE-AD and NP-AD group).

With the patients’ permission, a database was created using the files that were prepared by the procedure and that were dynamically tracked during pregnancy.

Patients older than eighteen years of age, those with gestational ages over twenty weeks, those with singleton pregnancies, pregnancies with risk factors for PE, those without other pregnancy problems, and those that provided informed consent were the selection criteria for PE patients. Clinically evident infections, severe liver and renal disease, diabetes, pregnancies with fetal congenital defects, multifetal pregnancies, hematological dysfunction, immunological diseases, and lack of informed permission were the exclusion criteria.

PE was defined according to The American College of Obstetricians and Gynecology criteria [56] and the International Society for the Study of Hypertension in Pregnancy (ISSHP) [57] criteria, including hypertension during pregnancy (usually systolic blood pressure (SBP) > 140 mm Hg and/or diastolic BP (DBP) > 90 mm Hg after 20 weeks of gestation) in a previously normotensive woman and resolving completely by the 6th postpartum week and significant proteinuria (at least 0.3 g/L in a 24 h collection of urine).

#### 2.2.1. Diagnosis of Hypertension

The first determination of BP was performed in a clinical setting (clinic or office, obstetric day unit, or in hospitalized patients). BP was measured using a standardized technique, the position of the pregnant woman was sitting, feet placed on the floor, with the cuff size adapted according to the circumference of the middle of the arm ≥ 33 cm, as well as the initial determination in both arms. BP was established based on an average of at least two measurements and confirmed after another determination at 15 min.

In patients with elevated BP, weekly out-of-office BP monitoring was recommended by means of home blood pressure monitoring with a clinically validated device for use in pregnancy and preeclampsia (recommended with the Microlife WatchBP Home S device), requiring that the patient self-measure their BP at least twice a day. Thus, pregnant women with PE were followed up to delivery.

For women with antepartum hypertension, BP was monitored at least once daily for the first 6 weeks after delivery, after which the physiological changes due to pregnancy subsided.

#### 2.2.2. Estimation of Proteinuria

Participants were advised to collect, early in the morning, 10–15 mL of urine in sealed sterile containers. Proteinuria was measured using a urine reagent dipstick via the semiquantitative color scale method. The semi-quantitative method with the strips helped us to classify proteinuria as absent, trace, or the presence of a quantity of 0.3 g/L, 1.0 g/L, or 3.0 g/L, corresponding to a negative result, the presence of traces, or a positive result assessed as 1+, 2+ and 3+, respectively. A positive test was considered to be ≥0.3 g/L (≥1+).

#### 2.2.3. Obtaining and Processing of Blood Samples

The first samples were collected at visit 2, after which, three months after delivery, the second samples (visit 3) were collected. Each pregnant woman had a 5 mL venous blood sample drawn in the morning using basic vacutainers (Becton Dickinson vacutainer). No later than 4 h after harvesting, the clot was separated by means of centrifugation (3000× *g* for 10 min), following a standardized protocol.

To facilitate longer sample processing, the serum sample containers were coded for each patient, sealed to prevent contamination, and kept at below −80 °C. To accommodate and prevent freezing–unfreezing cycle activities, the frozen samples were left at room temperature before dealing with the patient samples.

### 2.3. Immunological Assessment

For the assessment of serum levels of 8-iso-PGF2α, TNF-α, IL-6, IL-10, hs-CRP, and PTX3, the Enzyme-Linked Immunosorbent Assay (ELISA) technique was used, in the Immunology Laboratory of the University of Medicine and Pharmacy of Craiova.

We used commercial test sets designed specifically for each of the mediators, which were as follows: 8-iso-PGF2α (Cat.No.:E-EL-0041; sensitivity: 9.38 pg/mL; detection range: 15.63–1000 pg/mL), TNF-α (Cat.No.:E-EL-H0109; sensitivity: 4.69 pg/mL; detection range: 7.81–500 pg/mL), IL-6 (Cat.No.:E-EL-H6156; sensitivity: 0.94 pg/mL; detection range: 1.56–100 pg/mL), IL-10 (Cat.No.:E-EL-H6154; sensitivity: 0.94 pg/mL; detection range: 1.56–100 pg/mL), hs-CRP (Cat.No.:E-EL-H5134; sensitivity: 9.38 pg/mL; detection range: 15.63–1000 pg/mL), PTX3 (Cat.No.:E-EL-H6081; sensitivity: 4.12 pg/mL; detection range: 6.86–5000 pg/mL), Elabscience (Houston, TX, USA).

The recommended method and the manufacturer’s instructions were followed for the dilutions and working processes. A 450 nm wavelength standard optical analyzer was employed in the procedure.

### 2.4. Statistical Analysis

Data were analyzed using GraphPad Prism 5 Trial Version (LLC, San Diego, CA, USA). Data normality was tested by using the D’Agostino and Pearson omnibus normality test. The 8-iso-PGF2α, TNF-α, IL-6, IL-10, hs-CRP, and PTX3 had a normal distribution and were presented as mean values with the standard deviation (SD). The categorical values were expressed as percentages.

The difference between the groups was assessed using one-way ANOVA for parametric variables and the Kruskal–Wallis test for non-parametric variables. The existence of significant correlations between the levels of the 8-iso-PGF2α and inflammatory status biomarkers (TNF-α, IL-6, IL-10, hs-CRP, and PTX3) was evaluated using Pearson’s coefficients (−1 < r < 1), and the correlation heatmap matrix, whose colors vary from vivid red for strong negative correlations to bright green for strong positive correlations, was used to visually represent the results. A *p*-value of less than 0.05 was considered statistically significant.

The diagnostic values of studied markers were evaluated using ROC curve analysis. The performance was expressed as the area under the ROC curve (AUC) and p statistics for the difference between the calculated AUC and AUC = 0.5 (weak discriminative marker). We identified cut-off values that yielded the best accuracy, and we computed the sensitivity, specificity, and Youden index (sensitivity + specificity − 1) for each marker for the different threshold values that were examined.

## 3. Results

### 3.1. Biological Parameters of the Study Subjects

Our study highlighted that there was no statistically significant difference regarding maternal age (mean ± SD) between the PE and NP groups (29.00 ± 7.30 vs. 27.00 ± 4.40 years, *p* = 0.200). Gestational age (mean ± SD) at the time of PE diagnosis confirmation did not differ statistically significantly compared to the NP group (25.00 ± 2.12 vs. 25.67 ± 1.69 weeks, *p* = 0.217).

Between the two groups, we observed statistically significant differences regarding the other parameters measured at recruitment: maternal weight (83.45 ± 7.97 vs. 77.51 ± 7.22 kg, *p* = 0.020), BMI (33.00 ± 5.90 vs. 24.00 ± 2.20 kg/m^2^, *p* < 0.0001), SBP (170.00 ± 15.00 vs. 140.00 ± 7.40 mmHg, *p* = 0.0001), DBP (100.00 ± 15.00 vs. 83.00 ± 7.80 mmHg, *p* = 0.042), and MAP (120.00 ± 13.00 vs. 110.00 ± 10.00 mmHg, *p* = 0.0081).

Our analysis observed that the other indices assessing PE, namely aspartate aminotransferase (AST), alanine aminotransferase (ALT), uric acid, creatinine, and urea, showed statistically significant differences between PE patients and women with NP. We outlined the most significant clinical information for each of the two research groups in Table 1. Also, proteinuria was found in 18 (60.00%) patients in the PE group.

### 3.2. Patients with PE Exhibited Significantly Elevated Levels of 8-Iso-Prostaglandin F2alpha

The 8-iso-PGF2α values were higher in the PE and PE-AD groups than in the NP group (8.34- and 3.64-fold, respectively) (Table 2, Figure 1). Analyzing whether there is a decrease in serum values of 8-epi-PGF2 α in the NP group after delivery (NP-AD), we found a 1.41-fold decrease in serum values, but not statistically significant.

In our study, it was observed that in the PE-AD group, the serum values of 8-iso-PGF2α decreased significantly, by 2.29-fold compared to the PE group.

### 3.3. Inflammatory Status Biomarkers in the Study Groups

The PE group’s evaluated mediator values were found to be the highest, considerably higher than those of the other groups (Table 3).

Significantly higher levels of every evaluated inflammatory mediator (TNF-α, IL-6, IL-10, hs-CRP, and PTX3) were seen in both of the preeclamptic patient groups (PE + PE-AD) as compared to the healthy pregnant women group (NP) (Figure 2A–E).

### 3.4. 8-iso-PGF2α Levels Associated Positively with Inflammatory Mediators (TNF-α, IL-6, IL-10, hs-CRP, and PTX3)

The findings showed a strong positive correlation between the SBP and the 8-iso-PGF2α levels (strong correlation, r = 0.622, *p*-value = 0.020) and DBP (moderate correlation, r = 0.584, *p*-value = 0.036) (Figure 3).

Significant positive correlations with 8-iso-PGF2α levels were found for TNF-α (strong correlation, r = 0.645, *p*-value = 0.002), IL-10 (moderate correlation, r = 0.432, *p*-value = 0.045) and PTX3 (moderate correlation, r = 0.547, *p*-value = 0.038).

Other significant positive correlations were found between the PTX3 levels and the SBP (moderate correlation, r = 0.438, *p*-value = 0.016) and DBP (moderate correlation, r = 0.356, *p*-value = 0.029).

Significantly negative correlations were found between TNF-α and SBP (moderate correlation, r= −0.450, *p*-value = 0.038) and between IL-10 and hs-CRP (moderate correlation, r= −0.427, *p*-value = 0.038).

We also investigated whether the serum 8-iso-PGF2α and cytokine levels of the PE participants were related to their laboratory parameters. We noticed the following significant values: serum 8-iso-PGF2α levels were weakly but significantly positively correlated with the urea (r = 0.410, *p*-value = 0.036), AST (r = 0.345, *p*-value = 0.021), and ALT (r = 0.326, *p*-value = 0.048). Also, among the inflammatory markers, only PTX3 was correlated weakly but statistically significantly with creatinine (r = 0.485, *p*-value = 0.031), and ALT (r= 0.388, *p*-value = 0.028).

### 3.5. Diagnostic Accuracy of the Biomarkers

Using the ROC curve, we aimed to evaluate the diagnostic accuracy of the parameters (8-iso-PGF2α, TNF-α, IL-6, IL-10, hs-CRP, PTX3, SBP, and DBP) when diagnosing pregnant women with PE.

Analyzing the ROC curves obtained for the eight evaluated parameters, we noticed that the most accurate diagnosis of pregnant women with PE was obtained with SBP (99.30% accuracy) and DBP (98.80% accuracy), followed by 8-iso-PGF2α (94.90% accuracy) and PTX3 (73.30%) (Table 4). Consequently, when evaluating the predictive power of the examined indicators, we highlight the SBP’s diagnostic performance (Figure 4). Also, it can be observed that IL-6 values have good diagnostic accuracy (91.30%), slightly lower than that of PTX3. Our study revealed in the case of the other cytokines, TNF-α and IL-10, a much lower diagnostic accuracy (78.80% accuracy, and 75.20%, respectively).

The prediction of pregnant women with PE using the ROC curve showed the following significant values (Table 4): the cut-off value of 8-iso-PGF2α was found to be >230.80, with 93.33% sensitivity, 96.67% specificity, and a Youden index of 0.900 (AUC = 0.949, *p* < 0.0001); the cut-off value of PTX3 was found to be >4.315, with 93.33% sensitivity, 80.00% specificity, and a Youden index of 0.733 (AUC = 0.930, *p* < 0.0001); the cut-off value of IL-6 was found to be >46.52, with 83.33% sensitivity, 80.00% specificity, and a Youden index of 0.633 (AUC = 0.913, *p* < 0.0001) (Figure 5).

## 4. Discussion

We investigated oxidative stress by assessing lipid peroxidation and oxidative damage as indicated by 8-iso-PGF2α and biomarkers of inflammatory status with the measurement of the serum levels of TNF-α, IL-6, IL-10, hs-CRP, and PTX3 in preeclampsia during pregnancy and three months after delivery, in a group of newly diagnosed and untreated pregnant women with PE. We observed increased serum levels of 8-iso-PGF2α, PTX3, IL-6, hs-CRP, TNF-α, and IL-10 when the group of pregnant women with PE was included in the study, parameters which displayed a significant decrease in their expression when we re-evaluated the pregnant women three months after delivery. Thus, our hypothesis that the expression of these parameters decreases to a level comparable to the normal level three months after delivery was confirmed to be valid.

We began by selecting the research topic based on the observation that there is insufficient information on the role of oxidative stress development as detected by F2-isoprostanes (8-iso-PGF2α) in PE, regarding the follow-up of women after delivery, as well as the supposed association between serum levels of 8-iso-PGF2α and biomarkers of inflammatory status, such as TNF-α, IL-6, IL-10, hs-CRP, and PTX3.

Normal pregnancy induces a systemic inflammatory response, and the endothelium actively participates in the inflammatory network. It is known that the endothelium activates leukocytes and vice versa, while cytokines participate in signal transmission in this interaction. Increased circulating levels of IL-6 and TNF-α [58] were determined during pregnancy compared to nonpregnancy. Cytokines also act on the vascular tone and permeability, and they modulate the activity of cyclooxygenase (COX) with the production of mediators, such as prostaglandin E2 and prostaglandin F2α (PGE2, PGF2α), in different cells and induce the synthesis of PTX3 [59], a protein with an active role in innate immunity and reproduction. PGF2α mainly has vasoconstrictive and proinflammatory properties, with higher levels of PGF2α being observed in pregnant than in non-pregnant women [50]. Also, the classic acute-phase reactant, CRP, shows a modest increase in pregnancy [60], just like its family of proteins of PTX3 origin.

To avoid the fetal allograft being rejected, a state of systemic and local inflammation is induced via the combination of all the above-mentioned factors. Together with oxidative stress and angiogenesis, inflammatory factors are crucial to the healthy development of pregnancy. As a result, it is clear that oxidative stress and chronic inflammation are related, and we can think of them as an unbreakable pair.

In preeclampsia, the local and systemic inflammatory response, oxidative stress, changes in angiogenic factors, and vascular reactivity, are exacerbated, setting up compensatory mechanisms, and ultimately leading to placental and vascular dysfunction [61,62].

Some studies have found that F2-isoprostanes are considered the best available biomarkers of oxidative stress status and lipid peroxidation in in vivo dysfunction [63,64], and it was observed that other isoprostanes, such as D2- and E2-isoprostanes, are less suitable, as they are less stable. Roberts et al. [63] highlighted that F2-isoprostane measurement has several advantages over other quantitative markers of oxidative stress, because they are chemically stable, are formed in vivo, are present in detectable amounts in all tissues and normal biological fluids, thus allowing the definition of a normal range, and significantly increased levels were obtained in animal models of oxidative damage. Also, the studies of Gopaul et al. [65,66] showed that F2-isoprostane levels are not affected by the dietary lipid content.

In a 2000 study, McKinney et al. [67] measured plasma, urinary, and salivary 8-isoprostane levels in pregnant women with preeclampsia, normotensive pregnant women, and nonpregnant women. They observed that free 8-isoprostane concentrations are increased in the plasma of women with severe preeclampsia. They did not measure F2-isoprostanes in the postpartum period. In another study, in women who were at low risk of preeclampsia, leptin, the plasminogen activator inhibitor-1/-2 ratio, and 8-epi-prostaglandin F(2alpha) were found to be elevated by Chappell et al. [68]. The vitamin supplement group’s results for ascorbic acid, 8-epi-prostaglandin F(2alpha), leptin, and plasminogen activator inhibitor-1/-2 were comparable to those of low-risk women.

During normal pregnancy, Palm et al. [69] highlighted that increased levels of 8-iso-PGF2α were successively associated with advancing gestational age. The authors reported for the first time that the high levels of F2-isoprostanes observed in late pregnancy decrease to basal levels in the postpartum period.

In our study, the 8-iso-PGF2α values were higher in the PE and PE-AD groups than in the NP group. We observed that in the postpartum period (three months after delivery), in the PE-AD group, the serum values of 8-iso-PGF2α decreased significantly, 2.29-fold compared to the PE group. The data obtained by us are in agreement with those mentioned by Palm et al. Also, when analyzing whether there is a decrease in serum values of 8-epi-PGF2 α in the NP-AD group, we found a 1.41-fold decrease in serum values, but not statistically significant.

A biomarker of oxidative stress, 8-iso-PGF2α, is a significant isoprostane, whose levels were evaluated in the urine and plasma by Ishihara et al. [48]. Additionally, in 18 pre-eclamptic, 19 normal pregnancy, and 20 non-pregnant women, the inflammatory response was assessed using plasma and urinary 15-keto-dihydro-PGF2α, a significant cyclooxygenase-catalyzed PGF2α metabolite, as well as plasma α- and γ-tocopherol. PGF2α metabolite and 8-iso-PGF2α levels were found to be substantially greater in pregnant women than in non-pregnant women. Pre-eclamptic women’s levels of 8-iso-PGF2α were not different from those with a regular pregnancy, while the levels of PGF2α metabolites were much greater in women with a normal pregnancy. The authors concluded that higher levels of isoprostanes and prostaglandin metabolites at the end of pregnancy suggest the importance of both free radicals and oxidation products catalyzed by cyclooxygenase in the normal biological processes of pregnancy.

As in the case of other pathologies, animal models have proven to be useful research instruments for examining the preeclampsia pathophysiology, diagnostic standards, and therapeutic approaches. Shu et al. [70] sought to establish and evaluate a preeclampsia-like model in Sprague Dawley rats using N-nitro-L-arginine methyl ester (L-NAME). The plasma malondialdehyde and 8-iso-PFG2α levels were measured using quantitative sandwich ELISA kits. Every group treated with L-NAME had elevated plasma levels of 8-iso-PFG2α, and notable variations were noted when compared to the control group.

In a review by Drejza et al. [71], it was shown that during the analyzed period of 2012–2022, a lot of different markers of oxidative stress were used in the literature on obstetrics and gynecology, such as MDA, nitrous oxide (NO), total antioxidant capacity (TAC), total antioxidant activity (TAA), superoxide dismutase (SOD), glutathione peroxidase (GPx), glutathione peroxidase (4 GPx), glutathione reductase (GR), lipid peroxidation (LPO), 8-hydroxydeoxyguanosine (8-OHdG), oxidized glutathione (GSSG), catalase (CAT), superoxide (O_2_^−^), paraoxonase (PON-1), oxidative stress index (OSI), hs-CRP, 8-iso-PGF2α, PGF2α, glutathione (GSH), and glutathione transferase (GST).

Regarding the correlation of oxidative stress and inflammation during pregnancy, Eick et al. [72] suggested that oxidative stress and inflammation, as measured by 8-iso-PGF2α, its primary metabolite, and PGF2α biomarkers, may be important contributors to preterm birth (PTB; gestational age < 37 weeks). Eick’s meta-analysis [73] revealed a correlation between elevated odds of PTB, namely PTB of spontaneous origin and delivery before 34 full weeks of gestation, and oxidative stress as indicated by 8-iso-prostaglandin-F2α, F2-IsoP-M, and prostaglandin-F2α levels in urine.

Understanding the pathophysiology of PE, as well as the risks it can generate, is important in preventing maternal/fetal morbidity and mortality in the short and long term. The spectrum of these adverse results includes the cardiovascular system, renal pathologies, hepatic dysfunction, and damage to the nervous system [74]. It was found that eclamptic seizures are the most well-known neurological sequelae of preeclampsia [75].

Our study showed that 8-iso-PGF2α levels correlated very well with non-specific clinical signs such as hypertension, displaying a strong correlation with SBP, and a moderate correlation with DBP. The 8-iso-PGF2α levels correlated strongly and significantly with some inflammatory markers, such as TNF-α, IL-10, and PTX3. Our data showed significant correlations between serum 8-iso-PGF2α levels and laboratory parameters (urea, AST, and ALT). Also, among the inflammatory markers, only PTX3 correlated weakly but statistically significantly with creatinine and ALT.

Our findings demonstrate that we used the diagnostic efficacy of the SBP and DBP measures to assess the predictive value of the markers under investigation. Comparing the ROC curves for the eight parameters analyzed (8-iso-PGF2α, TNF-α, IL-6, IL-10, hs-CRP, PTX3, SBP, and DBP), we noticed that the most accurate diagnosis of pregnant women with PE was obtained with SBP and DBP, followed by 8-iso-PGF2α and PTX3. Also, it can be observed that IL-6 values have good diagnostic accuracy, slightly lower than that of PTX3. Our study revealed that the other cytokines, TNF-α and IL-10, provided a much lower diagnostic accuracy (78.80% accuracy, and 75.20%, respectively).

For 8-iso-PGF2α, the threshold value was >230.80 pg/mL, with 93.33% sensitivity, 96.67% specificity, and a Youden index of 0.900, in the diagnosis of pregnant women with PE. The ROC curve shows for PTX3 a threshold value of >4.315, with 93.33% sensitivity and 80.00% specificity, and a Youden index of 0.733.

As far as we are aware, this is the first investigation to assess 8-iso-PGF2α’s predictive utility. According to our results, although 8-iso-PGF2α and PTX3 had the same sensitivity (93.33%), 8-iso-PGF2α had a better specificity (96.67%) in the diagnosis of pregnant women with PE.

The differences between the study groups (PE, PE-AD, and NP) regarding the increased values of the 8-iso-PGF2α in the PE group, their considerable decrease after delivery, the fact that in the NP-AD group, no considerable decrease in the level of 8-iso-PGF2α was observed, the strong correlation between the values of this marker and SBP values, as well as the fact that 8-iso-PGF2α had a better specificity in the diagnosis of pregnant women with PE are results that suggest that the measurement of the serum levels of 8-iso-PGF2α could represent a prophylactic way to help clinicians identify or predict which pregnant women will develop PE.

We acknowledge that this study has some limitations because it was only conducted in our reference center and because of its descriptive character and limited sample size. The results of our investigation indicate a possible relationship between inflammatory biomarkers and the blood levels of 8-iso-PGF2α, but more research with multicenter participation would be beneficial to confirm these findings. Further investigations on this subject may yield valuable insights into identifying serum biomarkers that could be useful as supplementary treatment targets in clinical settings.

## 5. Conclusions

Our preliminary study showed that the expression level of serum 8-iso-PGF2α in PE patients was significantly higher than in the postpartum period (three months after delivery) in the PE-AD group, and the expression of 8-iso-PGF2α was positively correlated with hypertension and inflammatory markers. Our study has demonstrated that the 8-iso-PGF2α and PTX3 levels have the greatest diagnostic value for pregnant women with PE. 8-iso-PGF2α and PTX3 can be used as independent predictor factors, along with already-known cytokines, potentially representing a prophylactic way to help clinicians identify or predict which pregnant women will develop PE.

## Figures and Tables

**Figure 1 life-13-02242-f001:**
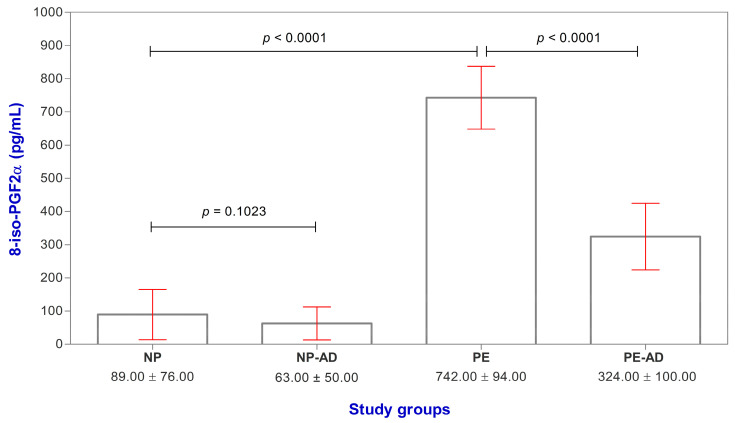
Serum levels of 8-iso-PGF2α (pg/mL) in the study groups; bars represent serum levels of 8-iso-PGF2α from individual samples; red horizontal lines represent the standard deviation.

**Figure 2 life-13-02242-f002:**
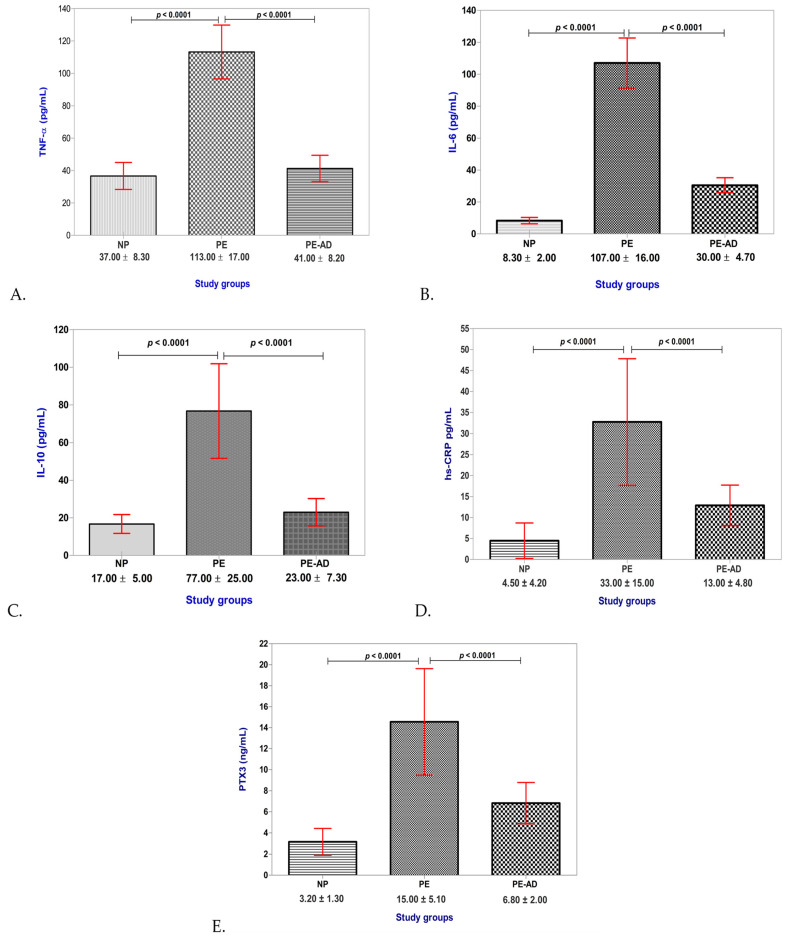
Serum levels of (**A**) TNF-α (pg/mL), (**B**) IL-6 (pg/mL), (**C**) IL-10 (pg/mL), (**D**) hs-CRP (pg/mL), and (**E**) PTX3 (ng/mL) in the study groups; bars represent serum levels of 8-iso-PGF2α from individual samples; red horizontal lines represent the standard deviation.

**Figure 3 life-13-02242-f003:**
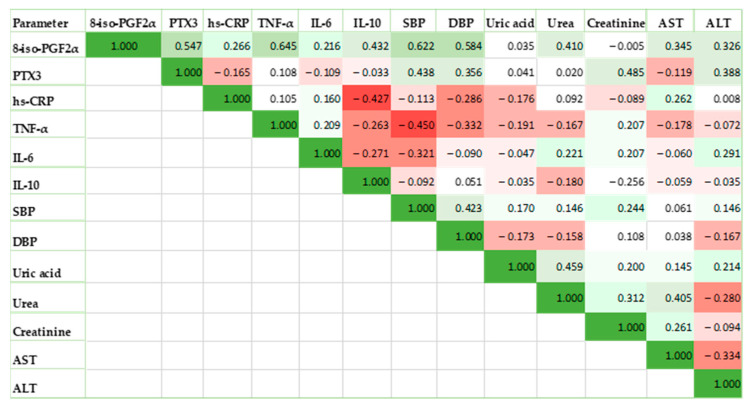
Measured indicators’ correlation heatmap for the PE group, where high positive correlations are shown in bright green and strong negative correlations are shown in brilliant red.

**Figure 4 life-13-02242-f004:**
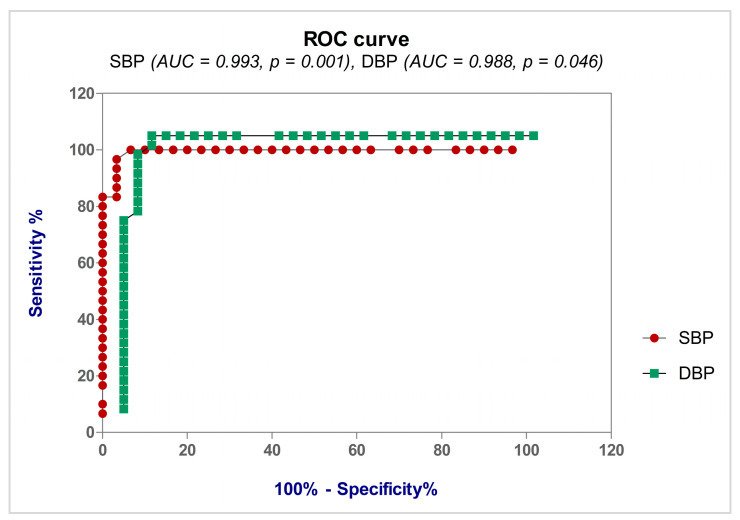
Receiver operating characteristic (ROC) curve for SBP and DBP.

**Figure 5 life-13-02242-f005:**
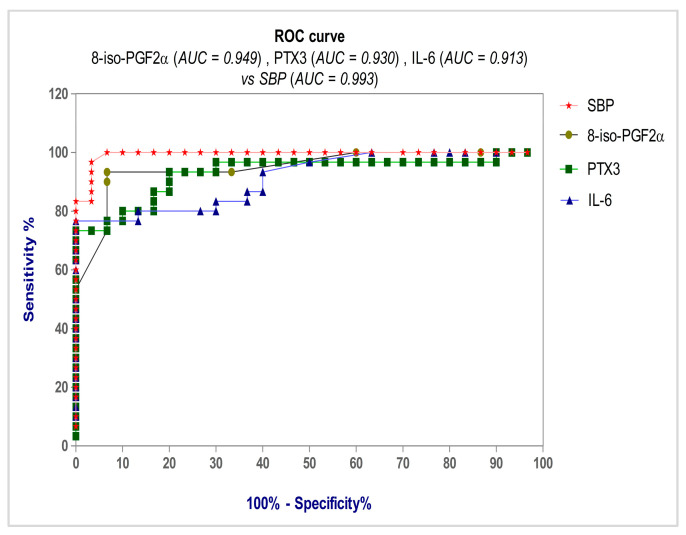
Receiver operating characteristic (ROC) curve for 8-iso-PGF2α, PTX3, and IL-6 vs. SBP.

**Table 1 life-13-02242-t001:** The enrolled patients’ clinical features and demographics.

Parameter	PE (*n* = 30)	NP (*n* = 30)	*p*-Value
Maternal age [years] (mean ± SD)	29.00 ± 7.30	27.00 ± 4.4	0.200
Maternal weight [kg] (mean ± SD)	83.45 ± 7.97	77.51 ± 7.22	0.020
BMI [kg/m^2^] (mean ± SD)	33.00 ± 5.90	24.00 ± 2.20	<0.0001
SBP [mmHg] (mean ± SD)	170.00 ± 15.00	140.00 ± 7.40	0.0001
DBP [mmHg] (mean ± SD)	100.00 ± 15.00	83.00 ± 7.80	0.042
MAP [mmHg] (mean ± SD)	120.00 ± 13.00	110.00 ± 10.00	0.0081
Diuresis:			
▪ normal	7	30	
▪ oliguria	5	–	
▪ proteinuria	18	–	
Hemoglobin [g/dL] (mean ± SD)	12.00 ± 1.39	12.00 ± 1.12	0.7264
Platelets [No. × 10^3^/μL] (mean ± SD)	160,000 ± 53,000	170,000 ± 31,000	0.3628
AST [U/L] (mean ± SD)	60.00 ± 23.00	49.00 ± 44.00	0.0451
ALT [U/L] (mean ± SD)	62.00 ± 38.00	39.0 ± 27.00	0.0035
Uric acid [mg/dL] (mean ± SD)	8.10 ± 0.50	5.80 ± 1.19	<0.0001
Creatinine [mg/dL] (mean ± SD)	0.82 ± 0.11	0.68 ± 0.11	<0.0001
Urea [mg/dL] (mean ± SD)	30.00 ± 8.57	21.00 ± 1.25	0.0001

ALT: alanine aminotransferase; AST: aspartate aminotransferase; BMI: body mass index; DBP: diastolic blood pressure; MAP: mean arterial blood pressure; NP: normal pregnant; PE: preeclampsia; SBP: systolic blood pressure; SD: standard deviation.

**Table 2 life-13-02242-t002:** Comparisons between the groups regarding 8-iso-PGF2α.

Parameter	PE (*n* = 30)	PE-AD (*n* = 30)	NP (*n* = 30)	NP-AD (*n* = 30)	*p*-Value from Kruskal–Wallis/One-Way ANOVA
8-iso-PGF2α (pg/mL) (mean ± SD)	742.00 ± 94.00	324.00 ± 100.00	89.00 ± 76.00	63.00 ± 50.00	*p* < 0.05

**Table 3 life-13-02242-t003:** Comparisons between the groups regarding inflammatory status biomarkers.

Parameter (Mean ± SD)	PE (*n* = 30)	PE-AD (*n* = 30)	NP (*n* = 30)	*p*-Value from Kruskal–Wallis/One-Way Anova
TNF-α (pg/mL)	113.00 ± 17.00	41.00 ± 8.20	37.00 ± 8.30	*p* < 0.05
IL-6 (pg/mL)	107.00 ± 16.00	30.00 ± 4.70	8.30 ± 2.00	*p* < 0.05
IL-10 (pg/mL)	77.00 ± 25.00	23.00 ± 7.30	17.00 ± 5.00	*p* < 0.05
hs-CRP (pg/mL)	33.00 ± 15.00	13.00 ± 4.80	4.50 ± 4.20	*p* < 0.05
PTX3 (ng/mL)	15.00 ± 5.10	6.80 ± 2.00	3.20 ± 1.30	*p* < 0.05

**Table 4 life-13-02242-t004:** Performance of the studied parameters in terms of diagnosis.

Parameter	AUC	Cut Off Values	Sensitivity %	Specificity %	Youden Index	*p*-Value
SBP	0.993	159.50	96.67	100.00	0.967	0.001
DBP	0.988	98.50	96.67	86.67	0.833	0.046
8-iso-PGF2α	0.949	230.80	93.33	96.67	0.900	<0.0001
PTX3	0.930	4.315	93.33	80.00	0.733	<0.0001
IL-6	0.913	46.52	83.33	80.00	0.633	<0.0001
hs-CRP	0.789	649.50	67.39	66.67	0.341	<0.0001
TNF-α	0.788	38.37	70.00	66.67	0.367	0.0156
IL-10	0.752	18.69	70.00	66.67	0.367	0.001

## Data Availability

The data used to support the findings of this study are available from the corresponding author upon reasonable request.

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
