# Peer review of "Association between Serum 8-Iso-Prostaglandin F2α as an Oxidative Stress Marker and Immunological Markers in a Cohort of Preeclampsia Patients"

_life, 2023, doi:10.3390/life13122242_

Round 1

Reviewer 1 Report

Comments and Suggestions for Authors

Increased oxidative stress has already been demonstrated in preeclampsia. The data in the manuscript confirm this. Personally, I find the results very interesting. But I don't see how measuring 8-iso-PGF2α levels would help clinicians diagnose or treat preeclampsia. It's an interesting, exciting topic, but the authors need to articulate goals or conclusions that go beyond preeclampsia severity classification. This was recently updated by the ISSHP and is not changed by this manuscript. The method (ELISA) is expensive, according to the data, it is very sensitive, but it does not give new results, so it will probably not be used for routine testing.

- The authors should explain in more detail the necessity and potential benefits of determining 8-iso-PGF2α.

- Table 1 is redundant, not in tabular format. Numbered questions would be more appropriate.

- Did the participants give written consent to participate in the experiment?

We will make up for this in the description of the ethical license! If you didn't need it, write this down too!

Author Response

Dear Reviewer,

Thank you very much for taking the time to analyze our manuscript, as well as for your kind appreciation and valuable suggestions.

All the typing recommended changes were performed in the body of our manuscript, with the Track Changes function activated.

Comments and Suggestions for Authors

Increased oxidative stress has already been demonstrated in preeclampsia. The data in the manuscript confirm this. Personally, I find the results very interesting. But I don't see how measuring 8-iso-PGF2α levels would help clinicians diagnose or treat preeclampsia. It's an interesting, exciting topic, but the authors need to articulate goals or conclusions that go beyond preeclampsia severity classification. This was recently updated by the ISSHP and is not changed by this manuscript. The method (ELISA) is expensive, according to the data, it is very sensitive, but it does not give new results, so it will probably not be used for routine testing.

- The authors should explain in more detail the necessity and potential benefits of determining 8-iso-PGF2α.

  • We formulated in line 461-468 (tracking changes without display marks): why it could be used for routine tests.

- Table 1 is redundant, not in tabular format. Numbered questions would be more appropriate.

  • Revised

- Did the participants give written consent to participate in the experiment? We will make up for this in the description of the ethical license! If you didn't need it, write this down too!

  • We specified in lines 138-139

Reviewer 2 Report

Comments and Suggestions for Authors

Dear authors, considering that the diagnostic criteria of preeclampsia (PE), among others, include proteinuria ≥0.3 g in a 24-hour urine specimen, platelet count <100,000/microL, serum creatinine >1.1 mg/dL and liver transaminases at least twice the upper limit of the normal, please describe in more detail the exact meaning of your findings and the exact meaning of “a better picture of the PE patients” in clinical practice.

Author Response

Dear Reviewer,

Thank you very much for taking the time to analyze our manuscript, as well as for your kind appreciation and valuable suggestions.

All the typing recommended changes were performed in the body of our manuscript, with the Track Changes function activated.

Comments and Suggestions for Authors

Dear authors, considering that the diagnostic criteria of preeclampsia (PE), among others, include proteinuria ≥0.3 g in a 24-hour urine specimen, platelet count <100,000/microL, serum creatinine >1.1 mg/dL and liver transaminases at least twice the upper limit of the normal, please describe in more detail the exact meaning of your findings and the exact meaning of “a better picture of the PE patients” in clinical practice.

  • We formulated in lines 461-468, why it could be used for routine tests, and the necessity and potential benefits of determining 8-iso-PGF2α.

Reviewer 3 Report

Comments and Suggestions for Authors

Dear authors,

congratulations, your paper is absolutely of interest

PE is an actual topic and unfortunately still a leading cause of maternal death therefore every effort in the direction of the study of such entity should be encourages

i would like to suggest few minor revisions

1) please add few sentences regarding PE severe complications (read and cite PMID: 35317697)

2) please double check spaces between words

MEthods are clear

Results clearly presented

discussion interesting

conclusions clear

best regards

Author Response

Dear Reviewer,

Thank you very much for taking the time to analyze our manuscript, as well as for your kind appreciation and valuable suggestions.

All the typing recommended changes were performed in the body of our manuscript, with the Track Changes function activated.

Comments and Suggestions for Authors

Dear authors,

congratulations, your paper is absolutely of interest

PE is an actual topic and unfortunately still a leading cause of maternal death therefore every effort in the direction of the study of such entity should be encourages

I would like to suggest few minor revisions

  • please add few sentences regarding PE severe complications (read and cite PMID: 35317697)
  • We revised according to the suggestions – line 433-437

  • please double check spaces between words
  • We checked

Reviewer 4 Report

Comments and Suggestions for Authors

The authors present a longitudinal, prospective cohort study on the difference in the maternal serum levels of inflammatory status markers and oxidative stress analytes between PE and non-complicated singleton pregnancies during and after gestation. The literature is not covering this field at all, and I would like to recommend the manuscript for publication with major revision. I think that the topic is interesting for the public of the journal.

The abstract has to be reorganized according to the following:

The Background section is a bit long. It is understandable that if there are many markers measured, then the correlation among the markers can be calculated. This fact is self-explanatory and should be excluded from the abstract in order to give place to other things. The statistics and the number of recruited patients are omitted in methods, which should be given. There are no numerical results expressed. The conclusion has to be more precise. What exactly recommend the authors? Are these factors predictors and can be used as a preceding markers of PE or the greater levels of these markers are just characteristic for this pregnancy pathology?

The MS text:

The ’Introduction’ section is generally too long. The importance of PE is not necessary to be emphasized in form of prevalence rates and effect on maternal mortality. The definition or the criteria of the PE are belonging rather to the Methods section and not the Introduction. The lines between 60 and 65 can be briefed. To interpret that assisted reproductive technology with donor sperm is increasing the risk of PE is beyond the authors scope and not necessary here. Otherwise, the sentence in the line 79 is not formed correctly. The content between the lines 93 and 95 can combine with the substance between the lines 60 and 65. The detailed explanation of the free radicals is not necessary. The overview of the inappropriate immune response, lipidperoxidation and immune markers are adequate.  Lines 119 and 120: I think it is not relevant to mention that PTX3 deficiency is associated with mice infertility.

There are too many research questions, and they are speculative. The illustration of the research questions should be simpler: interrelation of the markers and evaluation whether they are able to discriminate between the PE and normal patients. It is unnecessary and blatant to enlist the factors several times (in the introduction, in the research questions and in the Materials and methods) and repeating that in different context saying the same thing. The list of the factors is enough to be provided in the Materials and metods section.

Line 133: Do not need to say asked for permission from the local Ethical Committee, just the authors get it. The GDPR guidelines should be excluded from the Study design. It is self-explanatory and it is included in the Helsinki and Oviedo agreements. The study design is also verbose with redundancies. It is understood that the pregnant women were followed up from the 20th gestational week up to delivery in order to identify PE-cases and to distinguish them from the controls, but it is not written whether the mothers were checked weekly and how (a systematic BP measurement with urinary dips or how)? How they recruited exactly the patients: do the authors recruited first PE-patients thereafter the first norm control after the patient? There is no data provided on which gestational week do they started recruitment or were the controls matched in age or parity to the PE-patients?

How were the samples stored?

The set up of the database is also prolonging the MS and repeating that the pregnant women were monitored. The discrepancy between the mild PE and other cases of PE is irrelevant from the point of view of the study. Personally, I think that PE diagnostic criteria could be written in 3 lines and does not need to be so detailed (lines 171-184). However, the rest of the Materials and methods are good.

In the Results section the authors are not noticing in which gestational age were the PE and NP groups recruited. Were BMI registered at booking? How were the measurements of the blood pressure done (the values are at recruitment)? The comment relating the severity of the PE is based on the level of the liver and kidney function is not fully true. The severity of the PE can be denoted by when the symptoms appear and how intensive they are. So I would like to reword the sentence which starts in the line 229.

The authors stated in the Materials and methods that they excluded the comorbidities (lines 165-170), like diabetes and so on, but in the table 1 they highlight that they recruited PE-patients with renal pathology, diabetes and chronic hypertension. If they were not excluded the patients with these pathologic conditions may worsen the markers in the maternal serum.

When the PE appeared in the case cohort? In which gestational age were the maternal samples taken?

The other parts of the results section are good and sufficient. The results are as expected. The discussion is also adequate, and the authors are addressed their answer to their research questions.

Minor remarks:

Acronyms that are written only once in the MS, does not need to be spelled out. IVF or ICSI.

Preeclampsia should be marked as PE for the first time and thereafter does not needed to write out (like in the line 171).

Were there any trial to convert the data to a normal range of data with logaritmization

Generally, I think it is more appropriate to compare the three subgroups with Kruskal Wallis test to search for the tendency in the higher marker levels in the inflammatoric PE both in antepartum and postpartum period compared to the controls. However, it would be interesting to publish whether is there any decrease in these markers after partus in normal controls as well or not. Split-plot Anova methods can be used in these cases but normality is necessary to perform Anova. However, normality can be achieved by logarithmic, cubic or any other transformation of the data.

Author Response

Dear Reviewer,

Thank you very much for taking the time to analyze our manuscript, as well as for your kind appreciation and valuable suggestions.

All the typing recommended changes were performed in the body of our manuscript, with the Track Changes function activated.

Comments and Suggestions for Authors

The authors present a longitudinal, prospective cohort study on the difference in the maternal serum levels of inflammatory status markers and oxidative stress analytes between PE and non-complicated singleton pregnancies during and after gestation. The literature is not covering this field at all, and I would like to recommend the manuscript for publication with major revision. I think that the topic is interesting for the public of the journal.

The abstract has to be reorganized according to the following:

The Background section is a bit long. It is understandable that if there are many markers measured, then the correlation among the markers can be calculated. This fact is self-explanatory and should be excluded from the abstract in order to give place to other things. The statistics and the number of recruited patients are omitted in methods, which should be given. There are no numerical results expressed. The conclusion has to be more precise. What exactly recommend the authors? Are these factors predictors and can be used as a preceding markers of PE or the greater levels of these markers are just characteristic for this pregnancy pathology?

  • Revised

The MS text:

The ’Introduction’ section is generally too long. The importance of PE is not necessary to be emphasized in form of prevalence rates and effect on maternal mortality.

  • Revised

The definition or the criteria of the PE are belonging rather to the Methods section and not the Introduction.

  • Revised

The lines between 60 and 65 can be briefed. The content between the lines 93 and 95 can combine with the substance between the lines 60 and 65.

  • Revised

To interpret that assisted reproductive technology with donor sperm is increasing the risk of PE is beyond the authors scope and not necessary here. Otherwise, the sentence in the line 79 is not formed correctly.

  • Revised

The detailed explanation of the free radicals is not necessary.

  • Revised

The overview of the inappropriate immune response, lipidperoxidation and immune markers are adequate. 

Lines 119 and 120: I think it is not relevant to mention that PTX3 deficiency is associated with mice infertility.

  • Revised

There are too many research questions, and they are speculative. The illustration of the research questions should be simpler: interrelation of the markers and evaluation whether they are able to discriminate between the PE and normal patients. It is unnecessary and blatant to enlist the factors several times (in the introduction, in the research questions and in the Materials and methods) and repeating that in different context saying the same thing. The list of the factors is enough to be provided in the Materials and metods section.

  • Revised

Line 133: Do not need to say asked for permission from the local Ethical Committee, just the authors get it. The GDPR guidelines should be excluded from the Study design. It is self-explanatory and it is included in the Helsinki and Oviedo agreements.

  • Revised

The study design is also verbose with redundancies. It is understood that the pregnant women were followed up from the 20th gestational week up to delivery in order to identify PE-cases and to distinguish them from the controls, but it is not written whether the mothers were checked weekly and how (a systematic BP measurement with urinary dips or how)?

  • We introduced the subsection Diagnosis of hypertension - where we presented how BP was determined and monitored

How they recruited exactly the patients: do the authors recruited first PE-patients thereafter the first norm control after the patient?

  • We specified in lines 146-149 (tracking changes without display marks): The pregnant women were diagnosed consecutively in the Department of Gynecology of the Filantropia Municipal Clinical Hospital, Craiova, Dolj County, Romania, between September 2022 and March 2023. In total, sixty participants were comprised, divided into three study groups....

There is no data provided on which gestational week do they started recruitment or were the controls matched in age or parity to the PE-patients?

  • We specified in lines 181-183

How were the samples stored?

  • We specified in lines 193-203 : We introduced the subsection Collection and processing of blood samples

The set up of the database is also prolonging the MS and repeating that the pregnant women were monitored. The discrepancy between the mild PE and other cases of PE is irrelevant from the point of view of the study. Personally, I think that PE diagnostic criteria could be written in 3 lines and does not need to be so detailed (lines 171-184).

  • We removed lines 172-184 and revised them according to the suggestions

However, the rest of the Materials and methods are good.

In the Results section the authors are not noticing in which gestational age were the PE and NP groups recruited.

  • We specified in lines 242-244

Were BMI registered at booking? How were the measurements of the blood pressure done (the values are at recruitment)?

  • We specified in lines 245-246

The comment relating the severity of the PE is based on the level of the liver and kidney function is not fully true. The severity of the PE can be denoted by when the symptoms appear and how intensive they are. So I would like to reword the sentence which starts in the line 229.

  • Revised

The authors stated in the Materials and methods that they excluded the comorbidities (lines 165-170), like diabetes and so on, but in the table 1 they highlight that they recruited PE-patients with renal pathology, diabetes and chronic hypertension. If they were not excluded the patients with these pathologic conditions may worsen the markers in the maternal serum.

  • Revised

When the PE appeared in the case cohort? In which gestational age were the maternal samples taken?

  • We specified in lines 194-196

The other parts of the results section are good and sufficient. The results are as expected. The discussion is also adequate, and the authors are addressed their answer to their research questions.

Minor remarks:

Acronyms that are written only once in the MS, does not need to be spelled out. IVF or ICSI.

  • Revised

Preeclampsia should be marked as PE for the first time and thereafter does not needed to write out (like in the line 171).

  • Revised

Were there any trial to convert the data to a normal range of data with logaritmization

  • We did not try to convert the data into a normal range of data with logarithmization, because after testing the normal distribution of the data, using "D'Agostino & Pearson omnibus normality test" or "Shapiro-Wilk normality test", from GraphPad, "Passed normality test (alpha=0.05)" values.

Generally, I think it is more appropriate to compare the three subgroups with Kruskal Wallis test to search for the tendency in the higher marker levels in the inflammatoric PE both in antepartum and postpartum period compared to the controls. Split-plot Anova methods can be used in these cases but normality is necessary to perform Anova. However, normality can be achieved by logarithmic, cubic or any other transformation of the data.

  • We specified in the Statistical analysis section that we used the Kruskal Wallis test to compare the three subgroups.

However, it would be interesting to publish whether is there any decrease in these markers after partus in normal controls as well or not.

  • We I entered in Table 1 the values of the NP-AD group

Round 2

Reviewer 4 Report

Comments and Suggestions for Authors

The authors addressed well the questios, however, some small, minor revisions are required to publish the last version.

The minor remarks are the following:

Abstract:  What is PE-AD means?

Materials and methods: were there any who became attrited?

„Patient selection”: The PE and Non-PE groups were significantly different from age, so they were not matched in age. The process of recruiting controls are missing. Did the authors collected one PE patient thereafter the next potential (healthy) control or how? So did the authors recruit a PE-patient and thereafter a next healthy pregnant woman who had no any other diseases, and thereafter followed them up both the controls and PE-patients and did not come any gestation-related diseases (like gestational diabetes mellitus, intrahepatic cholestas) and so on. Or how did they collect the patients and controls? I assume that the inclusion criteria (lines 153-161) were the same for both groups, but it is not written so.

How much was the attrition rate (so the patient quit from the study? How many of the patients were excluded due to appearing complications that may affect the ROS, inflammatory and oxydative stress level?

In the statistical section some parts became confused:

The Kruskal-Wallis test is to those groups where the continuous variable is NOT FOLLOWING THE GAUSS CURVE. If the study characteristics DO FOLLOW THE NORMALITY, then ONE WAY ANOVA is the correct statistical probe. Now, in the MS is written in an opposite way.

If the factor levels are following the normality then SPEARMAN RANK CORRELATION is NOT adequate to calculate but different form of ANOVA.

Another advice: If one factor level gives a normality with Shapiro-Wilk’s test (or even with Kolmogorov-Smirnow test) then Pearson r are necessary, but if another factor level is not following the normality then Spearman’s rho is necessary.

It is not clarified in the MS.

Lines between 241-243: Gestational age (mean ± SD) at the time of PE diagnosis confirmation, did not 241 register statistically significant differences comparable to the NP group (25.00 ± 2.12 vs 242 25.67 ± 1.69 weeks, p=0.217).

Should be rewritten:

Gestational age (mean ± SD) at the time of PE diagnosis confirmation did not differ statistically significantly compared to the NP group (25.00 ± 2.12 vs 242 25.67 ± 1.69 weeks, p=0.217).

It is appreciated that the PE-group is very comparable with the healthy controls.

Application of Kruskal-Wallis test (if the variable is non-parametric) or Anova (parametric):

PE

NP

PE-AD

P-value from Kruskal-Wallis/One-way Anova

8-iso-PGF2α (pg/mL)

(mean±SD)

742.00 ± 94.00

324.00 ± 100.00

89.00 ± 76.00

P<0.05

It is not necessary, and by the way statistically not optimal to compare the part-results with each other. Namely, PE compared with NP is p<0.05 and thereafter PE compared with PE-AD is also p<0.05 and then NP is different frpm PE-AD resulting in significant result p<0.05. Instead of comparisons, it is better to make one single comparison with ANOVA/Kruskal-Wallis.

Another construction of the data:

PE

NP

Split-plot anova

8-iso-PGF2α (pg/mL)

(mean±SD)

742.00 ± 94.00

324.00 ± 100.00

PE-AD

NP-AD (n=30)

89.00 ± 76.00

63.00 ± 50.00

Here the simple and correct coparison is Split-plot Anova – repeat measurement Anova (with case-control comparison). Here is also one p-value will be the results.

But Anova can be used only the continuous variables that follow the normality.

(Sorry that it is not scientific, but there are even statistical you tube videos from reliable professors and scientific academies which interpret these).

Thereafter the authors can analyse the tendency.

The authors did not take away the Figure 1, interpret the tendency on a beautiful way. I mean the differences are given between the subgroups and I think to calculate a repeated anova and it is enough in the table 1.

Table 2 would be beneficial if they are calculating the Kruskal-Wallis or Anova and put only one p-values in each line. The authors made the separate analysis in the figures 2 A-E which is nice and enough.

In other words: both table 1 and figure 1 carries all p-values, but I recommend to put a Split-plot Anova p-value in the table since there are detailed p-values (between the sub-groups) on the top of the charts in the figure 1. The same is recommended to table 2 (p-value for Kruskal-Wallis/one way Anova) since the detailed p-values (between the sub-groups) or in the Figure 2.

Remember: Spearman rho, Kruskal-wallis is for continuous variables with no Gaussian distribution, whereas Anova, Split-plot Anova and Pearson rho is for variables with adequate normality.

If the authors have possibility and they have data then they should publish the NP-AD data in the postpartum period as well. Very beautiful article and conclusion can be written.I think the article is important and insteresting providing new tools in PE prognosis.

Author Response

Dear Reviewer,

Thank you very much for taking the time to analyze our manuscript, as well as for your kind appreciation and valuable suggestions.

All the typing recommended changes were performed in the body of our manuscript, with the Track Changes function activated.

Comments and Suggestions for Authors

The authors addressed well the questios, however, some small, minor revisions are required to publish the last version.

The minor remarks are the following:

Abstract:  What is PE-AD means?

  • revised

Materials and methods: were there any who became attrited?

„Patient selection”: The PE and Non-PE groups were significantly different from age, so they were not matched in age. The process of recruiting controls are missing. Did the authors collected one PE patient thereafter the next potential (healthy) control or how? So did the authors recruit a PE-patient and thereafter a next healthy pregnant woman who had no any other diseases, and thereafter followed them up both the controls and PE-patients and did not come any gestation-related diseases (like gestational diabetes mellitus, intrahepatic cholestas) and so on. Or how did they collect the patients and controls? I assume that the inclusion criteria (lines 153-161) were the same for both groups, but it is not written so.

  • revised

How much was the attrition rate (so the patient quit from the study? How many of the patients were excluded due to appearing complications that may affect the ROS, inflammatory and oxydative stress level?

  • revised

In the statistical section some parts became confused:

The Kruskal-Wallis test is to those groups where the continuous variable is NOT FOLLOWING THE GAUSS CURVE. If the study characteristics DO FOLLOW THE NORMALITY, then ONE WAY ANOVA is the correct statistical probe. Now, in the MS is written in an opposite way.

If the factor levels are following the normality then SPEARMAN RANK CORRELATION is NOT adequate to calculate but different form of ANOVA.

Another advice: If one factor level gives a normality with Shapiro-Wilk’s test (or even with Kolmogorov-Smirnow test) then Pearson r are necessary, but if another factor level is not following the normality then Spearman’s rho is necessary.

It is not clarified in the MS.

  • We revised as suggested

Lines between 241-243: Gestational age (mean ± SD) at the time of PE diagnosis confirmation, did not 241 register statistically significant differences comparable to the NP group (25.00 ± 2.12 vs 242 25.67 ± 1.69 weeks, p=0.217).

Should be rewritten:

Gestational age (mean ± SD) at the time of PE diagnosis confirmation did not differ statistically significantly compared to the NP group (25.00 ± 2.12 vs 242 25.67 ± 1.69 weeks, p=0.217).

  • We revised as suggested

It is appreciated that the PE-group is very comparable with the healthy controls.

Application of Kruskal-Wallis test (if the variable is non-parametric) or Anova (parametric):

PE

NP

PE-AD

P-value from Kruskal-Wallis/One-way Anova

8-iso-PGF2α (pg/mL)

(mean±SD)

742.00 ± 94.00

324.00 ± 100.00

89.00 ± 76.00

P<0.05

It is not necessary, and by the way statistically not optimal to compare the part-results with each other. Namely, PE compared with NP is p<0.05 and thereafter PE compared with PE-AD is also p<0.05 and then NP is different frpm PE-AD resulting in significant result p<0.05. Instead of comparisons, it is better to make one single comparison with ANOVA/Kruskal-Wallis.

  • We revised as suggested

Another construction of the data:

PE

NP

Split-plot anova

8-iso-PGF2α (pg/mL)

(mean±SD)

742.00 ± 94.00

324.00 ± 100.00

PE-AD

NP-AD (n=30)

89.00 ± 76.00

63.00 ± 50.00

Here the simple and correct coparison is Split-plot Anova – repeat measurement Anova (with case-control comparison). Here is also one p-value will be the results.

But Anova can be used only the continuous variables that follow the normality.

(Sorry that it is not scientific, but there are even statistical you tube videos from reliable professors and scientific academies which interpret these).

Thereafter the authors can analyse the tendency.

The authors did not take away the Figure 1, interpret the tendency on a beautiful way. I mean the differences are given between the subgroups and I think to calculate a repeated anova and it is enough in the table 1.

  • We revised as suggested

Table 2 would be beneficial if they are calculating the Kruskal-Wallis or Anova and put only one p-values in each line. The authors made the separate analysis in the figures 2 A-E which is nice and enough.

In other words: both table 1 and figure 1 carries all p-values, but I recommend to put a Split-plot Anova p-value in the table since there are detailed p-values (between the sub-groups) on the top of the charts in the figure 1. The same is recommended to table 2 (p-value for Kruskal-Wallis/one way Anova) since the detailed p-values (between the sub-groups) or in the Figure 2.

  • We revised as suggested

Remember: Spearman rho, Kruskal-wallis is for continuous variables with no Gaussian distribution, whereas Anova, Split-plot Anova and Pearson rho is for variables with adequate normality.

 If the authors have possibility and they have data then they should publish the NP-AD data in the postpartum period as well. Very beautiful article and conclusion can be written.I think the article is important and insteresting providing new tools in PE prognosis.
